# Effects of vermicompost and lime on acidic soil properties and malt barley (*Hordeum Distichum L*.) productivity in Mecha district, northwest Ethiopia

**Zenebe Terefe**[1,2]*, **Tesfaye Feyisa**[3☯], **Eyayu Molla**[2☯], **Workineh Ejigu**[4☯]

**1** Department of Plant Science, Woreta College of Agriculture, Woreta, Ethiopia, **2** Department of Natural Resources Management, College of Agriculture and Environmental Sciences, Bahir Dar University, Bahir Dar, Ethiopia, **3** Department of Soil and Water, Amhara Regional Agricultural Research Institution, Bahir Dar, Ethiopia, **4** Department of Natural Resources Management, College of Agriculture and Environmental Sciences, Debre Tabor University, Debre Tabor, Ethiopia

☯ These authors contributed equally to this work.

\* zenebegebreyesus2008@gmail.com

**Data Availability Statement:** All relevant data are within the manuscript and its Supporting Information files.

## Abstract

Due to continuous cultivation, high soil acidity, and low nutrient inputs, soil fertility depletion has been a major threat to northwest Ethiopia's crop productivity and food security. This study aimed to examine the effects of vermicompost and lime rates on soil properties and malt barley (*Hordeum distichum L*.) productivity under a furrow irrigation system on acidic soil in the Mecha district, northwest Ethiopia. The treatments were combinations of three levels of vermicompost (0, 2.66-, and 5.31-tons ha$^{-1}$) and three levels of lime (0-, 2.16-, and 3.24-tons ha$^{-1}$) arranged in a randomized complete block design with three replications. The results showed that the integrated application of 5.31 tons of vermicompost and 3.24 tons of lime ha$^{-1}$ provided the highest soil pH (6.20), available phosphorus (8.55 mg kg$^{-1}$), total nitrogen (0.25%), and organic carbon (3.40%). On the other hand, adding vermicompost and lime in combination or separately noticeably decreased the exchangeable acidity and aluminum toxicity. Besides, the integrated application of 5.31 tons vermicompost (VC) and 3.24 tons lime (L ha$^{-1}$ provided the highest dry biomass (12.22 tons ha$^{-1}$), grain yield (5.30 tons ha$^{-1}$), and net benefit (197, 246 Ethiopian Birr (ETB). Overall, the integrated application of vermicompost and lime can substantially increase soil fertility and crop yields. However, this study needs further testing and validation at varied rates in other areas.

## 1. Introduction

Soil acidity and nutrient depletions are among the major constraints affecting sustainable crop production in the highlands of Ethiopia [1,2]. Much of the Ethiopian highlands have acidic soil and low phosphorus availability caused by heavy precipitation resulting in the leaching of basic cations [3]. It is estimated that 43% of the Ethiopian cultivated land is affected by soil

**Funding:** The author(s) received no specific funding for this work.

**Competing interests:** There are no any competing interest related to this research study.

acidity [4], and from this, 28.1% of these soils are strongly acidic (pH 4.1–5.5) in nature [5]. Thus, poor nutrient availability due to high soil acidity is one of the main constraints causing low crop yields [6]. In general, soils exposed to acidification problems determine the overall activity of microbial population and soil health, since soil microbial diversity is the core driver of nutrient cycling and soil health [7].

Malt Barley (*Hordeum distichum L.*) is an important crop cultivated by small-scale farmers in Ethiopia's highlands. Although barley is considered the fifth-most common, widely farmed crop in Ethiopia's highlands, its productivity is very low due to extreme soil acidity and inadequate nutrient inputs application [8]. Due to the growth of the malt and beer industries, malt barley is widely cultivated and covers an area of 950,742 hectares of land with a productivity of 2.52 tons ha$^{-1}$ in the highlands of Ethiopia [9]. In the Amhara region, the area coverage (32,515 hectares) and productivity of malt barley (2.3 tons ha$^{-1}$) are extremely low compared to the potential yield obtained from field experiments [10]. Besides, rain-fed agriculture alone cannot meet the demand for malt barley.

Liming is an effective method to increase soil pH, reduce the adverse effects of exchangeable acidity and aluminum toxicity, and thereby enhance available phosphorus, and crop yield [11,12]. Moreover, it improves the activity of microorganisms as well as the availability and use efficiency of nutrients [13]. However, the practice of liming by smallholder farmers is rare in Ethiopia due to its high cost and limited supply. Thus, there is a need to find viable strategies that can reduce the lime amount for smallholder farmers.

The application of vermicompost is another effective strategy to improve soil pH, reducing aluminum toxicity while decreasing the lime amount required [14]. Vermicompost increases soil microbial activity such as nitrogen-fixing and phosphorus-solubilizing organisms [15]. Besides, it provides several essential macro and micronutrients, enhances oxygen availability, regulates soil temperature, improves porosity and infiltration, and boosts crop yield and quality [16]. However, applying vermicompost alone cannot replenish depleted nutrients due to limited organic inputs, and bulkiness.

Integrated application of organic and mineral fertilizers has become ecologically sustainable, socially acceptable, and an economically viable approach to yield greater productivity and maintain soil fertility [17]. Besides, integrated nutrient management practices considerably minimize nutrient losses through leaching, runoff, volatilization, and immobilization while maximizing nutrient use efficiency [18]. However, the combined effects of vermicompost and lime on acidic soil properties and malt barley productivity under irrigation systems have not been well studied. Consequently, assessing the effects of vermicompost and lime on soil properties and crop yield is vital to developing economically feasible and environmentally friendly strategies that can increase soil quality and crop productivity. Therefore, the objective of this study was to evaluate the effect of vermicompost and lime application on soil properties and malt barley productivity in acidic soils of Mecha district, northwest Ethiopia.

## 2. Materials and methods

### 2.1. Description of the study area

The study was carried out in 2021 and 2022 cropping seasons using furrow irrigation at the Koga irrigation scheme of Mecha district, Northwest Ethiopia. The study area is located between 11˚ 29' 30" to 11˚ 33' 0" N latitudes and 37˚ 07' 0" to 37˚ 10' 30" E longitudes (Fig 1). Its elevation ranges between 1500 and 2400 meter above sea level. The agroecology of the study area is characterized as a cool, semi-humid agroecosystem with distinct dry and rainy seasons. The rainfall pattern is unimodal and lasts from May through October. Based on 29

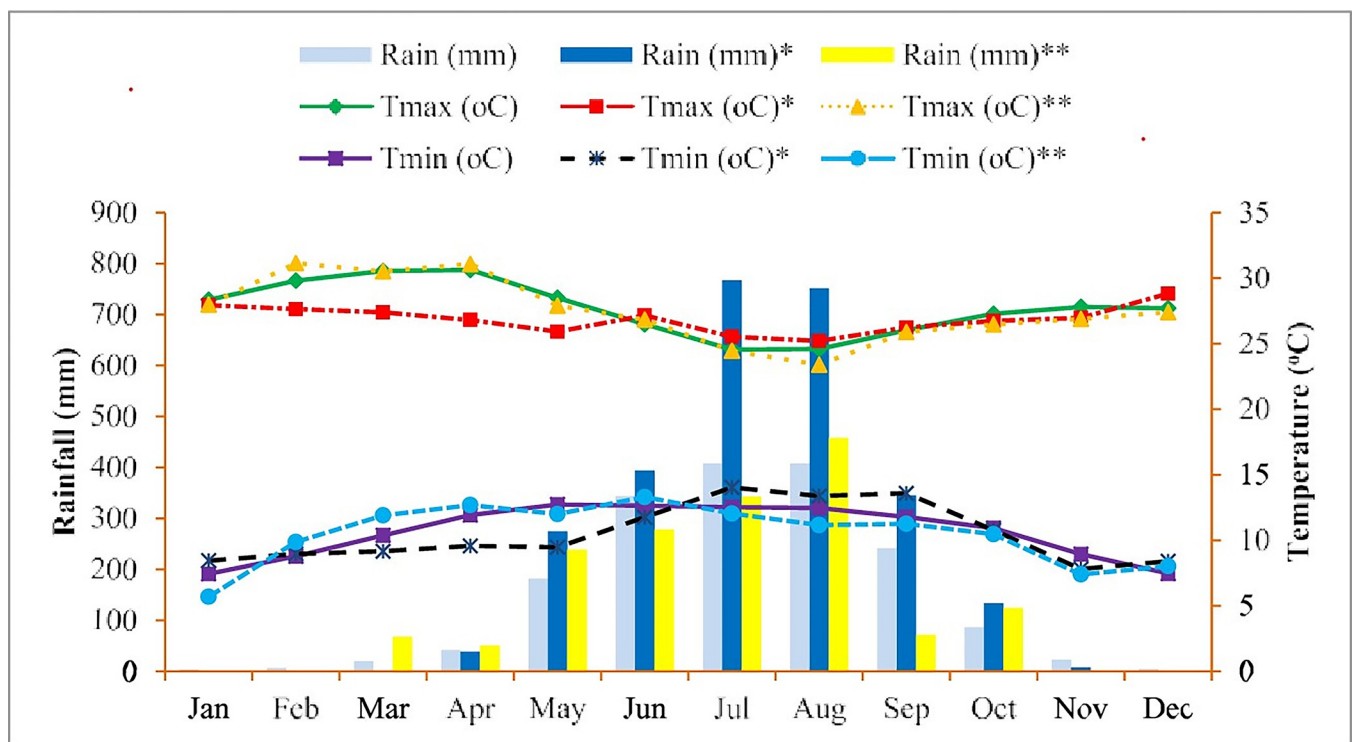

**Fig 1.** Average monthly rainfall (mm), maximum and minimum temperatures (˚C) of the study area over 29 years (1994–2022), during 2021(*) and 2022(**) (source: National Meteorological Agency of Ethiopia, Merawi Agro-meteorological station).

years (1994 to 2022) of climate data, the study area receives a mean annual rainfall of 1768 mm. The mean maximum and minimum temperatures are 27.7˚C and 10.7˚C, respectively. During the growing season (January to April), the mean monthly maximum temperatures in 2021 and 2022 growing seasons were 27.1˚C and 29.7˚C, while the minimum temperatures were 9.1˚C and 10.43˚C, respectively [Unpublished] (Fig 1). The major soil type of the experimental site is Nitisol, characterized by low pH and high exchangeable acidity [19]. The topography of the area exhibits distinct variation and the slope ranges from nearly flat to very steep, the nature of the topographical features has made it very liable to gully formation and extensive soil erosion. About 72% of local farmers is used for subsistence agriculture which includes crop production and animal husbandry in two types of agriculture i.e. rainfed & irrigated agriculture. The major types of crops grown in the vicinity include wheat, barley, maize, teff, and horticulture. Farmers widely apply inorganic fertilizers like urea and NPS (Nitrogen-Phosphorus-Sulfur blend fertilizer) to increase crop yield. Furthermore, furrow irrigation is a common irrigation practice in the study area [20].

## 2.2. Experimental materials and procedures

Before treatment application, the nutrient contents of vermicompost were evaluated to determine how much vermicompost should be applied. The recommended amount of mineral nitrogen fertilizer (69 kg N ha$^{-1}$) obtained from VC was applied for malt barley production in the study area. Similarly, the amount of lime needed was determined based on the exchangeable acidity method [21] (Eq 1). The limestone ($CaCO_3$) utilized in the experiment had high quality, with a 98% quality content and a fineness of 99%. Assuming that one mole of

exchangeable acidity would be neutralized by an equivalent mole of $CaCO_3$.

$$\text{LR}\left(CaCO_3 \Big| \frac{kg}{ha}\right) = \frac{\text{EA}\left(\frac{cmol}{kg \ of \ soil}\right) \times 0.15m \times 10^4 m^2 \times BD(g|cm^3) \times 1000}{2000} \times 1.5 \qquad (1)$$

Where LR = Lime Requirement, EA = Exchangeable acidity (cmol (+) kg$^{-1}$ of soil), 0.15 m = Soil Depth, $10^4$ m$^2$ = "Farm Size (ha), and BD = Bulk Density (1.20 g/cm$^3$), and 1.5 is the multiplication factor applied based on suggestion [21].

The recommended dose of lime and vermicompost was applied one month before sowing seeds and thoroughly incorporated into the soil through the broadcasting method. The technique employed for distributing irrigation water was furrow irrigation. Before seeding, the field was irrigated for three days to create an optimum soil condition for seeding. As a test crop, the IBON174/03 malt barley yield (*Hordeum distichum L.*) variety was sown at a seed rate of 100 kg ha$^{-1}$ in rows spacing with 20 cm. Four growth stages were used to split the entire growing period. The early stage, development stage, mid-season stage, and late-season stage are all included. The soil water content at field capacity (FC) and permanent wilting point (PWP) was determined by the pressure plate apparatus technique, whereas total available water (TAW) was obtained by subtracting PWP from FC. The field was watered to field capacity level every 7 days from the beginning until mid-season and every 15 days from the middle of the season until the first week of the late season. Irrigation ceased after the first week of the late season. All plots received the recommended rate of 60.50 kg ha$^{-1}$ phosphorus (NPS) fertilizer (19 N 38 $P_2O_5$ 7S) at sowing.

## 2.3. Treatments and experimental setup

The treatments consisted of factorial combinations of three levels of vermicompost (0-, 2.66-, and 5.31-tons ha$^{-1}$ based on nitrogen equivalency ratio) and three levels of lime (0-, 2.16-, and 3.24-tons ha$^{-1}$ based on exchangeable acidity method) in a randomized complete block design (RCBD) with three replications. The plot size was 2 m by 3 m (6 m$^2$) with spacing of 1 m and 1.5 m between plots and blocks, respectively. The treatment combinations are shown in Table 1.

**Table 1. Experimental treatments setup.**

| No. | Treatments | Amount of input applied (tons ha$^{-1}$) | |
| :---: | :---: | :---: | :---: |
| | | Vermicompost | Lime |
| 1 | VC0L0 | 0 | 0 |
| 2 | VC1L0 | 2.66 | 0 |
| 3 | VC2L0 | 5.31 | 0 |
| 4 | VC0L1 | 0 | 2.16 |
| 5 | VC0L2 | 0 | 3.24 |
| 6 | VC1L1 | 2.66 | 2.16 |
| 7 | VC1L2 | 2.66 | 3.24 |
| 8 | VC2L1 | 5.31 | 2.16 |
| 9 | VC2L2 | 5.31 | 3.24 |

Where, VC0L0 = without vermicompost and lime (control), VC1 = 2.66 tons ha$^{-1}$ vermicompost (50% Nitrogen equivalence), VC2 = 5.31 tons ha$^{-1}$ vermicompost (100% Nitrogen equivalence), L1 = 2.16 tons ha$^{-1}$ lime (100% lime requirement) and L2 = 3.24 tons ha$^{-1}$ lime (150% lime requirement). The lime requirement is based on exchangeable acidity.

## 2.4. Soil sampling and laboratory analysis

Before planting, soil samples were collected by auger from the corner and in the center of the experimental field at a depth of 0–20 cm. After harvest, soil samples were collected from five points in each treatment plot. The subsamples were properly mixed to create a composite soil sample for each treatment plot. The samples were brought to the laboratory after being labeled and placed in a plastic bag. The soil samples were air-dried, crushed, ground, and passed through a 2 mm sieve, except for total nitrogen and organic carbon, which further passed through a 0.5 mm sieve.

The soil texture was analyzed using the Bouyoucus hydrometer method [22]. The soil bulk density was determined using the core sampler method stated in [23]. Soil pH was measured potentiometrically in 1:2.5 soils with $H_2O$ using a combination glass electrode pH meter [24]. The cation exchange capacity (CEC) was analyzed at pH 7 using 1N ammonium acetate to remove the cations from the soil using the method described by [25]. The exchangeable acidity was determined by saturating the soil samples with 1M KCl solution [26]. Wet combustion was used to determine the soil organic carbon (SOC) content [27]. The total nitrogen (TN) was analyzed using the Kjeldahl method [28]. The available phosphorous (Av P) was extracted using the Bray-II method designed for acid soils [29]. The exchangeable basic cations (Ca, and Mg) were quantified using atomic absorption spectrophotometry, whereas exchangeable Na and K were measured using flame photometers [30].

## 2.5. Crop data collection

The following phenological data were collected, including days to 50% heading and days to 90% maturity. Ten randomly selected plants from the center row were used to record the yield and yield components of the malt barley, such as plant height, spike length, number of spikelets per spike, and number of seeds per spike. The averaged data were then used. For ten randomly collected plants from the middle rows of each plot, the plant height was measured and recorded starting from the soil surface to the tip of the spike. At maturity, the crop was removed from the net plot and sun-dried until a constant weight was reached. The above-ground dry biomass was then calculated for each plot and recorded in tons (t) per hectare. Manual separation of the grains from the straws provided the grain yield measurements for each plot in tons. The harvest index was calculated using equation (Eq 2) as follows:

$$\text{Harvest Index}(\%) = \frac{Grain\ Yield\left(\frac{t}{ha}\right)}{Above\ ground\ dry\ biomass\left(\frac{t}{ha}\right)} \times 100 \qquad (2)$$

From the grain samples taken at random from the net plots created for each treatment, the weight of a thousand grains was corrected to a seed moisture content of 10%. The yield of grains was calculated using a sensitive balance, 12.5% moisture adjustment, and conversion to hectare [31].

$$Grain\ Yield\left(\frac{t}{ha}\right) = Yield\ realized\left(\frac{t}{ha}\right) \times \frac{\text{Actual moisture content at } 100\% - 12.5}{100}$$

$$Above\ ground\ dry\ biomass\left(\frac{t}{ha}\right) \qquad (3)$$

## 2.6. Data analysis

Analysis of variance (ANOVA) was performed using Statistical Analysis System (SAS) software version 9.4 to examine the effects of treatments on soil and crop parameters. Mean separation was done using the least significant differences (LSD) test at 5% levels of significance. Moreover, the Pearson correlation between grain yield and related components was performed.

## 2.7. Economics analysis

Economic analysis was estimated based on the method developed by [32]. The partial budget analysis was calculated with a 10% grain yield adjustment since available evidence showed that the grain yields obtained from farmers' fields could be reduced by 10% compared to yields from experimental plots [32]. The total variable costs were calculated by adding the costs of lime, vermicompost, chemical fertilizers used, and other operating expenditures. Dominance analysis was performed to determine which options were the most cost-effective. To accomplish this, the treatments were set up in ascending order of total variable costs. The net benefit for each treatment was determined, the total variable costs of each treatment were subtracted from the gross net return for each. If the net benefit of one treatment was less than the net benefit of a treatment with lower total variable costs, the treatment with higher total variable costs was rejected and the other treatment was considered dominant. The minimum acceptable marginal rate of return was over 50% to 100% for a treatment to be regarded as a viable option for farmers [32].

## 2.8. Ethics statement

This study does not contain any experimental studies with human participants or animals and did not perform biological experiments using genetic materials (plants, seeds, germplasms).

# 3. Results and discussion

## 3.1. The nutrient contents of experimental soil and vermicompost before planting

The pre-planting nutrient contents of the soil and vermicompost is presented in Table 2. The soil was clay in texture, with an optimum value of bulk density (1.2 g/cm$^3$), which is suitable for agricultural activities. According to the ratings of [33], the soil's pH (5.08) was extremely acidic, with an exchangeable acidity of 1.60 cmol (+) kg$^{-1}$ and an exchangeable aluminum ($Al^{+3}$) of 0.96 cmol (+) kg$^{-1}$), The contents of soil organic carbon, total nitrogen, and available phosphorus were rated as low, while CEC and exchangeable bases ($Ca^{2+}$, $Mg^{2+}$, $K^+$, and $Na^+$) were found in the optimum ranges (Table 2). Overall, the properties of the study site's soil had poor fertility, and vital soil nutrients were not readily available.

The nutrient compositions of vermicompost used as an organic input for the experiment are shown in Table 2. The laboratory test result showed that the applied vermicompost to the experimental field was effectively mineralized for nutrients, enhancing their availability in the soil for plant use, as evidenced by the neutral pH and the low C: N ratio. It also had a high available phosphorous and cation exchange capacity. Similarly, [34] asserts that organic matter with a C: N ratio of less than 20 is likely to degrade and mineralize, and as a result, microorganisms may release and produce nutrients from vermicompost.

**Table 2. The nutrient contents of experimental soil and vermicompost before planting.**

| Soil | | | | Vermicompost | |
|---|---|---|---|---|---|
| Parameters | Values | Rating | References | Parameters | Values |
| pH (H$_2$O) | 5.08 | very strongly acidic | Landon (2014) | pH (H$_2$O) | 7.02 |
| SOC (%) | 2.12 | low | >> | TN (%) | 1.30 |
| TN (%) | 0.13 | low | | OC (%) | 6.85 |
| Av. P (mg kg$^{-1}$) | 5 | low | >> | OM (%) | 11.81 |
| CEC | 20 | medium | | Av. P | 85.20 |
| Ca | 4.9 | low | Hazelton & Murphy (2007) | CEC | 65 |
| Mg | 2.1 | moderate | >> | | |
| K | 0.4 | moderate | >> | | |
| Na | 0.22 | low | | | |
| Ex. Ac | 1.6 | high | | | |
| Ex. Al | 0.96 | High | | | |
| BD (g/cm$^3$) | 1.20 | medium | | | |
| Sand (%) | 28.16 | | | | |
| Clay (%) | 40.52 | Clay texture | | | |
| Silt (%) | 31.32 | | | | |
| FC (%) | 36 | | | | |
| PWP (%) | 24.2 | | | | |

pH = soil reaction; TN = total nitrogen; SOC = soil organic carbon; Av. P = available phosphorus; CEC = cation exchangeable capacity, Ca = exchangeable calcium, Mg = exchangeable magnesium, K = exchangeable potassium, Na = exchangeable sodium, BD = bulk density, FC = soil water content at field capacity, PWP = soil water content at permanent wilting point.

## 3.2. Interaction effects of vermicompost and lime on selected soil properties

**3.2.1. Soil pH, exchangeable acidity, and aluminum.** The result revealed that the integrated application of vermicompost and lime had a significant ($p<0.05$) effect on soil pH, exchangeable acidity, and aluminum contents. The soil pH improved from being strongly acidic (5.08) to slightly acidic (6.20) level. The highest soil pH (6.20) was recorded from the combined application of 5.31 tons VC + 3.24 tons L ha$^{-1}$. On the other hand, the combined application of vermicompost and lime decreased exchangeable acidity and aluminum (Table 3). The decline in H$^+$ and Al$^{3+}$ concentrations in the soil solution and the rise in soil pH could result from the combined neutralizing and buffering effects of lime and vermicompost. The separate application of vermicompost or lime affects soil pH. Similarly, [35] reported that the application of vermicompost alone resulted in a change in soil pH from 5.1 to 5.5 by 0.4 units. Since vermicompost has a high concentration of basic cations and pH, it can replace the acidic cations from the exchangeable sites. The primary reason for liming reducing soil acidity was because of its remarkable acid-neutralizing properties, capacity to eliminate preexisting acid, improvement of the bio-physicochemical properties of the soil, and reduction of toxicity of heavy metals [36]. Similarly, the addition of lime to organic matter has a positive influence on soil pH [37]. The mechanisms responsible for increase in soil pH could be the formation of organo-Al complexes that reduce Al$^{3+}$ toxicities or direct neutralization of Al$^{3+}$ caused by increased the soil organic matter [38]. Moreover, vermicompost and lime applied jointly significantly increased the soil pH and reduced the extent of exchangeable acidity [39]. [40] indicated that a combination of 7.5 tons ha$^{-1}$ vermicompost and 4 tons lime ha$^{-1}$ increased soil pH from 4.80 to 6.05 and decreased exchangeable acidity and Al from 2.38 to 0.17 cmolkg$^{-1}$ and 0.45 to 0.09 cmolkg$^{-1}$, respectively. Additionally, [35] the combination of lime with

**Table 3. Interaction effects of vermicompost and lime on selected soil properties.**

| Treatments | pH | Ex. Ac | Ex. Al | SOC | TN | Av. P | CEC | Ca | Mg |
|---|---|---|---|---|---|---|---|---|---|
| VC0L0 | 5.14[d] | 1.63[a] | 0.98[a] | 2.05[g] | 0.15[g] | 4.91[i] | 18.95[h] | 4.18[f] | 2.00[f] |
| VC1L0 | 5.21[cd] | 1.42[b] | 0.83[b] | 2.65[e] | 0.19[de] | 5.26[h] | 21.47[e] | 7.34[d] | 2.76[d] |
| VC2L0 | 5.28[cd] | 1.22[c] | 0.66[c] | 2.99[c] | 0.196[cd] | 6.08[e] | 22.21[d] | 8.01[c] | 3.06[c] |
| VC0L1 | 5.36[c] | 1.02[d] | 0.54[d] | 2.20[f] | 0.17[f] | 5.56[g] | 20.25[g] | 6.44[e] | 2.34[e] |
| VC0L2 | 5.76[b] | 0.91[e] | 0.53[d] | 2.26[f] | 0.18[ef] | 5.79[f] | 20.74[f] | 7.47[d] | 3.49[b] |
| VC1L1 | 5.84[b] | 0.82[e] | 0.48[de] | 2.86[d] | 0.21[c] | 7.53[d] | 23.22[c] | 8.10[bc] | 3.33[bc] |
| VC1L2 | 5.89[b] | 0.66[f] | 0.41[ef] | 3.19[b] | 0.22[b] | 7.90[c] | 23.35[c] | 8.12[bc] | 3.40[b] |
| VC2L1 | 5.93[b] | 0.54[g] | 0.37[fg] | 3.29[ab] | 0.24[a] | 8.36[b] | 24.54[b] | 8.43[b] | 3.45[b] |
| VC2 L2 | 6.20[a] | 0.44[g] | 0.30[g] | 3.40[a] | 0.25[a] | 8.55[a] | 25.09[a] | 9.10[a] | 4.04[a] |
| LSD (5%) | 0.18 | 0.10 | 0.08 | 0.12 | 0.01 | 0.14 | 0.28 | 0.42 | 0.28 |
| CV (%) | 1.87 | 6.12 | 7.97 | 2.48 | 3.86 | 1.23 | 0.74 | 3.25 | 5.24 |
| P-value | 0.008 | 0.572 | 0.095 | 0.005 | 0.052 | 0.0001 | 0.0001 | 0.001 | 0.0004 |
| Std Error (±) | 0.072 | 0.078 | 0.041 | 0.094 | 0.006 | 0.263 | 0.378 | 0.269 | 0.119 |

The same columns that have the same letters after them are not statistically significant. pH = soil reaction, Ex. Ac = exchangeable acidity (cmol (+) kg$^{-1}$), Ex. Al = exchangeable aluminum (cmol (+) kg$^{-1}$), TN = total nitrogen (%), SOC = soil organic carbon (%),Av. P = accessible phosphorus (mg kg$^{-1}$), CEC = cation exchange capacity (cmol (+) kg$^{-1}$), Ca = exchangeable calcium (cmol (+) kg$^{-1}$), and Mg = exchangeable magnesium (cmol (+) kg$^{-1}$), VC = vermicompost, L = lime.

vermicompost at higher rates (75% to 100% L with 2.5 to 5 tons VC ha$^{-1}$) increased the soil pH from 5.1 to 5.8.

**3.2.2. Soil organic carbon, total nitrogen, and available phosphorus.** The combined application of vermicompost and lime significantly (p<0.05) improved Av P), TN, and SOC contents (Table 3). The highest SOC was found from the application of 5.31 tons of VC and 3.24 tons of L ha$^{-1}$ which increased the SOC by 39.71% followed by the application of 5.31 tons of VC and 2.16 tons of L ha$^{-1}$ which improved organic carbon by 36.69% compared to the control. This showed that the application of vermicompost and lime, either separately or together, raised the amount of SOC, which in turn creates favorable conditions to improve the microbial population [40]. According to [41], lime applications increased the soil's microbial activity and sped up the rate at which organic matter decomposed in the soil. A similar study [42], reported the improvement of SOC following the application of lime and manure related to the improvements of soil conditions. It has a direct effect on improving soil pH, enhancing organic matter might rather play a great role in binding toxic elements of Cu, Mn, and Al and fixing materials mainly oxides of Fe, and Al in acidic soils thereby reducing phosphorous fixation and enhancing soil biological activities [43,44]. In line with our result, [41] also exhibited that the integrated use of 4 tons ha$^{-1}$ limes and 7.5 tons ha$^{-1}$ vermicompost provided the highest content of SOC (4.1%). This result agreed with the finding of [45] who reported the highest organic carbon content after the combined application of 5 tons ha$^{-1}$ compost with 1.5 and/or 3 tons ha$^{-1}$ lime.

The interplay between vermicompost and lime also had a considerable impact on total N. The highest TN (0.25%) was recorded from plots treated with 5.31 tons VC and 3.24 tons L ha$^{-1}$ followed by 0.24% from 5.31 tons VC + 2.16 tons L ha$^{-1}$ (Table 3). This showed that the mineralization of vermicompost and lime effects could be the reasons for the increase in total nitrogen following the application of vermicompost and lime. The positive effect of lime and organic matter from compost and/ or vermicompost in improving total nitrogen in acidic soils was also reported by [45–47]. Vermicompost and lime either separately or in combination can improve biological nitrogen fixation in acidic soils, and increase the net mineralization of

organic nitrogen [48,49]. [50] also reported that the highest total N (0.29%) was obtained when lime at a rate of 4 tonsha$^{-1}$ was applied in combination with 7.5 tons VC ha$^{-1}$.

Similarly, the interaction effect of vermicompost and lime significantly (P< 0.05) improved the content of available soil phosphorous. The highest amount of available phosphorous (8.55 mg kg$^{-1}$) was found from the combined use of 5.31 tons VC and 3.24 tons L ha$^{-1}$ (Table 3). The increase in accessible phosphorous may be attributable to lime's rapid effects on reducing soil acidity and the subsequent increase in phosphorous availability [51]. The combined application of vermicompost and lime improved the amount of available phosphorous because it released phosphorous that had been adsorbed by iron and aluminum in the soil solution, which improved phosphorus mineralization from organic matter. According to [41], some anions and organic acids created during the breakdown of vermicompost hinder phosphorus fixation and raise the concentration of phosphorous in soil solution, which may make it more accessible to plants. While doing so, it created situations that were more conducive to microbial activity, which led to the net mineralization of organic phosphorus in the soil. In line with our result according to [35], the highest available P (13.27 mg kg$^{-1}$) was recorded under the combined application of lime 75%- and 2.5-tons VC ha$^{-1}$ about 72.34% increment over the control. Similar to this, [45] reported that the application of lime and compost in combinations or alone significantly increased soil available phosphorous.

**3.2.3. Cation exchange capacity (CEC) and exchangeable bases (Ca$^{2+}$ and Mg$^{2+}$).** The interaction of vermicompost and lime showed a significant effect (P<0.05) on the contents of CEC and exchangeable bases (Ca$^{2+}$, Mg$^{2+}$). Plots treated with 5.31 tons VC and 3.24 tons L ha$^{-1}$ provided the highest values of CEC (25.09 (cmol (+) kg$^{-1}$), exchangeable Ca$^{2+}$ (9.10 (cmol (+) kg$^{-1}$), and Mg$^{2+}$ (4.04 (cmol (+) kg$^{-1}$) while the lowest values were from the control. A substantial difference had been recorded between the highest and lowest values of CEC, exchangeable Ca$^{2+}$, and Mg$^{2+}$ 24.51%, 54%, and 50.49%, respectively (Table 3). Similar to this, [52] stated that soil organic matter (humus) is part of negatively charged soil colloidal materials that account for cation exchange, enhanced directly attributed to improving the cation exchange capacity of experimental soils. Similarly, [53] showed that the addition of compost increased the soil cation exchangeable capacity, exchangeable bases, and organic matter. In line with the report of [54], an increase in soil pH by adding lime leads to the de-protonation of hydrogen ions from pH-dependent charge sites of soil and increases the cation exchangeable capacity of the soil.

The addition of vermicompost and lime, either alone or in combination, increased soil exchangeable bases when acidic soils were ameliorated. An increase in vermicompost application rates also increases soil exchangeable calcium and magnesium concentrations due to the alkalinity of the applied vermicompost. The increment of exchangeable Mg$^{2+}$ in the soil might be due to the high Mg$^{2+}$ content of the applied vermicompost and increments in soil pH. Lime contains calcium cations to exchange hydrogen ions on the exchange sites and anions like CO$_3^{-2}$ to neutralize the hydrogen ions released from the sites and hydrolyze aluminum species in the soil solution. [55] reported that application of lime to acid soils increased exchangeable bases (Ca$^{2+}$ and Mg$^{2+}$) ions due to the replacement of acidic cations such as Al$^{3+}$, H$^+$, Mn$^{2+}$, and Fe$^{2+}$ ions by basic cations in the soil solution. This result was in line with [56], who reported that the highest calcium content was recorded after the combined application of lime and coffee husk compost at the rates of 4.8- and 10-tons ha$^{-1}$, respectively. This could be attributed to the release of Ca$^{2+}$ ions from lime through its dissolution and mineralization of compost. The result was consistent with the finding of [57], who reported that increased Mg content after the application of lime and manure was attributed to increasing soil pH and reduced Al$^{3+}$ and H$^+$.

Table 4. Interaction effects of vermicompost and lime on malt barley yield and yield components.

| Treatments | PH | SL | NSPS | NGPS | GY | DBY | TGW | HI |
|---|---|---|---|---|---|---|---|---|
| VC0L0 | 46.00[f] | 5.81[f] | 8.17[e] | 14.57[f] | 1.54[f] | 6.13[d] | 34.30[g] | 25.16[e] |
| VC1L0 | 53.90[ef] | 6.86[ef] | 9.76[d] | 18.46[e] | 2.19[e] | 6.65[d] | 36.16[fg] | 32.85[d] |
| VC2L0 | 66.97[d] | 7.95[cde] | 11.00[cd] | 23.63[bc] | 3.86[c] | 10.53[bc] | 43.00[cd] | 36.67[c] |
| VC0L1 | 61.60[de] | 7.67[de] | 10.63[cd] | 22.13[cd] | 2.79[d] | 10.08[c] | 40.83[de] | 27.64[e] |
| VC0L2 | 56.23[ef] | 7.19[e] | 9.97[d] | 19.26[de] | 2.45[de] | 6.42[d] | 37.97[ef] | 38.20[c] |
| VC1L1 | 72.20[cd] | 8.57[bcd] | 11.93[c] | 25.27[bc] | 4.39[b] | 11.20[abc] | 44.00[bcd] | 39.22[bc] |
| VC1L2 | 79.06[bc] | 9.05[abc] | 14.06[b] | 26.53[ab] | 4.64[b] | 11.37[ab] | 45.06[bc] | 40.89[ab] |
| VC2L1 | 86.73[ab] | 9.71[ab] | 14.87[ab] | 28.63[a] | 5.08[a] | 11.78[a] | 46.70[ab] | 43.09[a] |
| VC2L2 | 91.47[a] | 10.19[a] | 15.63[a] | 29.63[a] | 5.30[a] | 12.22[a] | 48.53[a] | 43.40[a] |
| LSD (5%) | 10.72 | 1.34 | 1.31 | 3.19 | 0.37 | 1.14 | 3.42 | 2.60 |
| CV (%) | 9.16 | 9.60 | 6.46 | 8.03 | 5.98 | 6.91 | 4.77 | 4.17 |
| p-value | 0.0160 | 0.0807 | 0.0020 | 0.0006 | 0.0001 | 0.0001 | 0.0001 | 0.0001 |
| Std. Error (±) | 3.02 | 0.29 | 0.49 | 0.97 | 0.26 | 0.47 | 0.96 | 1.23 |

Means in the same column that is followed by the same letter are not statistically distinct at a 5% probability level. Note that PH = plant height (cm), SL = spike length (cm), NSPS = numbers of spikelets per spike, NGPS = number of grains per spike, GY = grain yield (tons ha$^{-1}$), DBY = dry biomass yield (tons ha$^{-1}$), TGW = thousand-grain weight in grams, and HI = harvest index in percent.

## 3.3. Interaction effects of vermicompost and lime on malt barley yield and yield components

The results showed that plant height, spike length, number of spikelets per spike, and number of grains per spike were significantly affected (P<0.05) by the combined application of vermicompost and lime (Table 4). The application of 5.31 tons VC + 3.24 tons L ha$^{-1}$ gave the maximum plant height (91.47 cm), spike length (10.19 cm), number of spikelets per spike (15.63), and number of grains per spike (29.63), while the minimum values were recorded from the control (Table 4). [58] found that vermicompost and lime applied together had a more substantial impact on plant height than alone. In a similar vein, compared to untreated soil, limed soil produced more seeds per plant. Additionally, bread wheat yield increased with increased the application of lime [59,60], and organic manure [47]. In line with this finding, [61] reported that the application of nitrogen levels gave significantly higher colocasia tuber yield. In agreement with this result, the combined application of lime and manure in acid soils has the potential to contribute to an overall increase in yields because it reduces exchangeable acidity and increases pH and soil fertility [62]. The findings of [63] demonstrated that the application of organic matter improves the beneficial effects of microorganisms in the soil environments as an indicator of soil quality and sustainability of the ecosystems. Moreover, SOM and TN are the two promising indicators for soil fertility and provide nutrients suitable for plant growth. Similarly, the residual effect of lime and organic manure significantly increased the yield, nutrient content, and nutrient uptake of wheat [37]. The increase in grain yield over control ranged from 10.14 to 54.38%. In another experiment, [64] reported that the combined application of 5 tons of manure and 2.2 tons of ha$^{-1}$ lime increased grain and straw yield by 279% and 187%, respectively, over the control.

## 3.4. The correlation coefficient among phenological, yield, and yield-related parameters

Grains per spike (r = 0.961), plant height (r = 0.953), number of spikelets per spike (r = 0.946), dry biomass yield (r = 0.930), spike length (r = 0.919), harvest index (r = 0.846), and thousand-

**Table 5. The correlation coefficient among phenological, yield, and yield-related parameters.**

| Parameters | DH | DM | PH | SL | NSPS | NGPS | GY | DBY | TGW |
|---|---|---|---|---|---|---|---|---|---|
| DM | .621** | | | | | | | | |
| PH | .315[ns] | .282[ns] | | | | | | | |
| SL | .239[ns] | .352[ns] | .946** | | | | | | |
| NSPS | .336[ns] | .331[ns] | .974** | .954** | | | | | |
| NGPS | .341[ns] | .336[ns] | .957** | .968** | .954** | | | | |
| GY | .401* | .395* | .953** | .919** | .946** | .961** | | | |
| DBY | .309[ns] | .304[ns] | .893** | .879** | .868** | .933** | .930** | | |
| TGW | -.076[ns] | .052[ns] | .567** | .586** | .595** | .546** | .499** | .464* | |
| HI | .439* | .393* | .788** | .744** | .801** | .773** | .846** | .608** | .403* |

Correlation is significant at the 0.01 level (**), the 0.05 level (*), and non-significance (ns), respectively. Where PH = plant height (cm), SL = spike length (cm), NSPS = number of spikelets per spike, NGPS = number of grains per spike, GY = grain yield (ton $ha^{-1}$), DBY = dry biomass yield (ton $ha^{-1}$), TGW = thousand-grain weight (g), and HI = harvest index percent.

grain weight (r = 0.499) had a positive and strong correlation with grain yield. The correlations with days to heading and physiological maturity were also favorable (r = 0.401* and r = 0.395*), respectively (Table 5). The yield and related components were positively correlated and had linear responses and significant contributions to the yields of barley.

## 3.5. Partial budget analysis of treatments on barley yield

The partial budget analysis revealed that the application of 5.31 tons of VC and 3.24 tons of L $ha^{-1}$ in combination provided the highest net benefit (197,245.88 ETB), followed by a combination of 5.31 tons of VC and 2.16 tons of L $ha^{-1}$ (189,947 ETB) while the control treatment provided the lowest net benefit (50,954 ETB) (Table 6). The finding was in line with [65], who recommended the combined application of 5 tons of manure and 2.2 tons of lime $ha^{-1}$, which is economically feasible to increase wheat yield, and residual soil phosphorus in the acidic soils of the Gozamin district. Thus, the combined application of 5.31 tons of VC and 3.24 tons of L $ha^{-1}$ provides the highest net benefit which makes smallholder farmers economically benefited compared to the control (50,954 ETB).

**Table 6. Partial budget analysis of treatments on barley yield.**

| Treatments | GY | Ad.GY | SY | Ad.SY | GNR | TVC | NB | MRR |
|---|---|---|---|---|---|---|---|---|
| VC0L0 | 1.54 | 1.38 | 4.59 | 4.14 | 85,682 | 34,728 | 50,954 | - |
| VC0L1 | 2.79 | 2.51 | 7.29 | 6.56 | 151,688 | 41,884 | 109,804 | 822 |
| VC1L0 | 2.19 | 1.97 | 4.46 | 4.02 | 114,497 | 45,074 | 69,423 | D |
| VC0L2 | 2.45 | 2.20 | 3.97 | 3.57 | 124,507 | 45,462 | 79,044 | D |
| VC1L1 | 4.38 | 3.94 | 6.82 | 6.14 | 221,721 | 52,230 | 169,490 | 577 |
| VC2L0 | 3.87 | 3.48 | 6.67 | 6.00 | 197,984 | 55,419 | 142,564 | D |
| VC1L2 | 4.64 | 4.18 | 6.73 | 6.05 | 233,016 | 55,808 | 177,208 | 216 |
| VC2L1 | 5.08 | 4.57 | 6.71 | 6.04 | 252,523 | 62,576 | 189,947 | 188 |
| VC2L2 | 5.30 | 4.77 | 6.92 | 6.23 | 263,400 | 66,154 | 197,246 | 204 |

GY = grain yield (tons $ha^{-1}$), Ad.GY = adjusted grain yield (tons $ha^{-1}$), SY = straw yield (tons $ha^{-1}$), Ad.SY = adjusted straw yield (tons $ha^{-1}$), GNR = gross net return, TVC = total variable cost, NB = net benefits, MRR = marginal rate of return (%),D = dominance, ETB = Ethiopian Birr (1$ = 48.13 ETB, average value for 2021 and 2022).

## 4. Conclusion

The results showed the application of vermicompost and lime in combination noticeably reduced exchangeable acidity and aluminum contents thereby increasing soil nutrient availability and crop yields. The highest contents of soil organic carbon (3.40%), total nitrogen (0.25%), accessible phosphorus (8.55 mg kg$^{-1}$) and CEC (25.09 cmol (+) kg$^{-1}$) were observed under plots treated with 5.31 tons VC and 3.24 tons L ha$^{-1}$ in combination while the lost values were from the control. Likewise, vermicompost and lime applied at a rate of 5.31 tons VC + 3.24 tons L ha$^{-1}$ provided the highest grain yield (5.30 tons ha$^{-1}$) and dry biomass yield (12.22 tons ha$^{-1}$) and net benefit (197, 246 ETB). In conclusion, the integrated application of 5.31 tons ha$^{-1}$ vermicompost and 3.24 tons ha$^{-1}$lime is a promising practice to increase soil fertility and crop yields for smallholder farmers. However, it is imperative to consider further study to evaluate the long-term effects of vermicompost and lime application on the bio-physicochemical characteristics of the soils. Moreover, this study needs to be further tested and assessed at varied rates in diverse agro-ecologies with significant issues with soil acidity.

## Supporting information

**S1 Fig. Treated and untreated Barley Crops at Maturity stage.**
(PDF)

**S2 Fig. Treated and untreated Barley Crops at Vegetative stage.**
(PDF)

## Acknowledgments

This research was supported by Bahir Dar University. We thank the local farmers for permitting us to perform the field trial on their cropland and for their cooperation during the study years. We are also grateful to the Ethiopian Meteorological Services Agency for sharing rainfall and temperature data. I thank the Amhara Regional Bureau of Agriculture and Woreta College of Agriculture for permitting me to pursue my Ph.D. study.

## Author Contributions

**Conceptualization:** Zenebe Terefe, Tesfaye Feyisa, Eyayu Molla, Workineh Ejigu.

**Data curation:** Zenebe Terefe, Tesfaye Feyisa, Eyayu Molla, Workineh Ejigu.

**Formal analysis:** Zenebe Terefe.

**Investigation:** Zenebe Terefe, Tesfaye Feyisa, Eyayu Molla, Workineh Ejigu.

**Methodology:** Zenebe Terefe.

**Project administration:** Zenebe Terefe, Tesfaye Feyisa, Eyayu Molla, Workineh Ejigu.

**Resources:** Zenebe Terefe.

**Software:** Zenebe Terefe.

**Supervision:** Zenebe Terefe, Tesfaye Feyisa, Eyayu Molla, Workineh Ejigu.

**Validation:** Zenebe Terefe, Tesfaye Feyisa, Eyayu Molla, Workineh Ejigu.

**Visualization:** Zenebe Terefe, Tesfaye Feyisa, Eyayu Molla, Workineh Ejigu.

**Writing – original draft:** Zenebe Terefe.

**Writing – review & editing:** Zenebe Terefe, Tesfaye Feyisa, Eyayu Molla, Workineh Ejigu.

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
