## [Decision Letter · Decision Letter 0]

31 May 2024

PONE-D-24-13572Effects of vermicompost and lime on soil properties and malt barley (Hordeum distichum L.) productivity on acidic soils of Mecha district, northwest EthiopiaPLOS ONE

Dear Dr. Gebreyesus,

Thank you for submitting your manuscript to PLOS ONE. After careful consideration, we feel that it has merit but does not fully meet PLOS ONE’s publication criteria as it currently stands. Therefore, we invite you to submit a revised version of the manuscript that addresses the points raised during the review process.

We look forward to receiving your revised manuscript.

Kind regards,

Ravinder Kumar, Ph.D.

Academic Editor

PLOS ONE

3. We note that your Data Availability Statement is currently as follows: [no]

5. We note that you have referenced

(4. Haile, H., Asefa, S., Regassa, A., Demssie, W., Kassie, K. & Gebrie, S. (2017). Extension manual for acid soil management (unpublished report). ATA (ATA), ed.), Addis Ababa, Ethiopia.

19. Bahir Dar Meteorological Stations (BDMS). (2020). Rainfall, minimum and maximum temperature data from 2011-2020 in Banja district Awi zone Ethiopian from Bahir Dar. Unpublished document.)

which has currently not yet been accepted for publication. Please remove this from your References and amend this to state in the body of your manuscript: (ie “Bewick et al. [Unpublished]”) as detailed online in our guide for authors

7. We note that Figure 1 in your submission contain [map/satellite] images which may be copyrighted. All PLOS content is published under the Creative Commons Attribution License (CC BY 4.0), which means that the manuscript, images, and Supporting Information files will be freely available online, and any third party is permitted to access, download, copy, distribute, and use these materials in any way, even commercially, with proper attribution. For these reasons, we cannot publish previously copyrighted maps or satellite images created using proprietary data, such as Google software (Google Maps, Street View, and Earth). For more information, see our copyright guidelines: http://journals.plos.org/plosone/s/licenses-and-copyright.

8. Please include your tables as part of your main manuscript and remove the individual files. Please note that supplementary tables (should remain/ be uploaded) as separate "supporting information" files

Reviewers' comments:

Reviewer's Responses to Questions

**Comments to the Author**

1. Is the manuscript technically sound, and do the data support the conclusions?

Reviewer #1: Yes

Reviewer #2: Partly

Reviewer #3: Yes

2. Has the statistical analysis been performed appropriately and rigorously? 

Reviewer #1: Yes

Reviewer #2: Yes

Reviewer #3: Yes

3. Have the authors made all data underlying the findings in their manuscript fully available?

Reviewer #1: Yes

Reviewer #2: Yes

Reviewer #3: Yes

4. Is the manuscript presented in an intelligible fashion and written in standard English?

Reviewer #1: Yes

Reviewer #2: No

Reviewer #3: Yes

5. Review Comments to the Author

Reviewer #1: I have gone through the exhaustive article on “Effects of vermicompost and lime on soil properties and malt barley (Hordeum distichum L.) productivity on acidic soils of Mecha district, northwest Ethiopia” In the article, the previous literature is well-documented by the authors. However, the following research articles are needed to be cited in the introduction and methodology section.

.

1. DOI: 10.1007/s11104-023-06016-4

2. DOI: 10.3390/microorganisms10102078

3. DOI: 10.4236/ojss.2018.82007

Reviewer #2: Title: Please replace the word “of’ with the word ‘in’ before Mecha

Abstract:

Line 16: replace the word “of’ with the word ‘in’ before Mecha

Line 18: make the word arrange in past form

Please write clearly, about tons and ton; Use the elaborate form of VC and L

Keywords: Use more appropriate words

Introduction:

In the first sentence, use a conjunction to complete the sentence

In the second sentence add an article before the word Ethiopian and leaching

In the third sentence, ‘acidic (pH 4.1-5.5) soils’ write in nature instead of soils.

The first sentence and last sentence of first para (line 33-38) is almost same. Make it in one sentence.

Line 39: add scientific name of crop.

“Although barley is considered to be the fifth-most common, a widely farmed crop in Ethiopia's highlands, it is extremely susceptible to soil acidity” the sentence is not clear, make it clear and simple.

Please remove the decimal and give a round figure (Line 42).

Always maintain the same unit

Line 47: Please use recent information

Line 48-49: Rewrite the sentence considering the article, tense, and punctuation

Line 53 please use recent data

Line 54-56. Check these sentence “Besides, vermicompost is enriched with soil microbiota such as nitrogen-fixing and phosphorus-solubilizing organisms and provides several essential macro and micronutrients become more available in VC [15]. It including the ability to increase soil microbial activity by enhancing oxygen availability, regulating soil temperature, improving porosity, infiltration, and nutrient content, growth promoter and boosting crop yield and quality” and write clearly following grammatical rules

Line 61: Add an article before the word economically

Line 64: Mention maximizing instead of maximize

Line 66: Write have instead of has

Materials and Methods:

Please check the sentence “The study site is located between 11°” 29' 30" N and 11° 33' 0" N latitudes and 37° 07' 0" E and 37° 10' 30" E” longitudes”

The study was carried out in 2021 and 2022, but the meterological data was given from 2011 to 2020. Please give the data only during the cropping season.

Convert the word productions to production

Please rewrite the sentence “The major crops type growing the vicinity are wheat, barley, maize, teff and various types of horticultures”

Line 101 and 103: Check the unit of bulk density

Line 114: Use sowing instead of planting as seed was used for crop production

Please mention the source of N and P fertilizer

Please explain whether boron fertilizer was used or not because it is related to grain formation

Line 127: convert the word ground to grinded

Line 133: Please read the sentence carefully

Follow the same pattern for writing unit

Manual separation of the grains from the straws resulted in yield measurements for each plot in tons (kg ha-1). Write the sentence clearly

Check the word ‘hectare’ throughout the manuscript

Line 148: Unify the unit

Tables:

For better understanding, use the short name of the treatment instead of the percent and rate of vermicompost and lime. Then elaborate on the short name of the treatment in the table footnote

Synchronize the treatments serials in table-1 with table-3,4 and 6

Mention the area for vermicompost and lime rate

Please solve the problem of t/ha, t ha-1, ton/ha, t ha-1, tons, and ton

Please check the sentence “Table 2. Characteristics the study soil and vermicompost before planting” and correct the sentence

Table-1: Explain, for treatment combination, why don’t use 50% sole lime whether use 50% sole vermicompost?; use 150% sole lime but why not 150% vermicompost? also use combination of 150% VC+100 L but why not 150% L+100% VC

Table-2: BD is appearing very low, whether soil is clay and organic carbon content is very low

Table-3: Check the p vale of available Phosphorus

Table-6: Please use a round figure for NB and MRR, mentioned GNR in the table caption whether used GNB in the table footnote, write correct one

Unify the tables heading with the description heading in the result and discussion section

Please use the same font for footnote

Result and discussion

Please add the field capacity data as irrigation was supplied based on the field capacity

183-184 line: C:N and C/N , write this in a uniform pattern

Please remove clumsiness and duplication during the writing of results and discussion of results. Discuss the results specifically. Omit unnecessary results.

Please reduce the number of references and use recent references, make space for two or more references in the third bracket

Make similarity for writing p, ap, phosphorus, and available phosphorus

Line 264: Place a verb after the reference number 43

Line 276: Put a comma before respectively

Line 303: Remove the bracket before 29.63

Please add the value of the control treatment during the comparison

Partial budget analysis:

First sentence (line 326-329), please read the sentence carefully and rewrite the sentence for better understanding

Line 330: Please use a round figure for data value

Please check the net benefit value described in the abstract, partial budget analysis, and conclusion chapters and make uniform

Conclusion:

Consider the same pattern for RL, LR, and L

Line 344: lime compost? not clear

References:

References were written whimsically. Rewrite the reference chapter and follow a pattern for all reference.

Overall comments:

English of the manuscript is poor. Improvement is required. Please go through many rounds of editing, taking into account typos and grammatical problems, before submitting the updated work.

Following the addition of the aforementioned remarks, the manuscript can be approved.

Line 148: Unify the unit

Tables:

For better understanding, use the short name of the treatment instead of the percent and rate of vermicompost and lime. Then elaborate on the short name of the treatment in the table footnote

Synchronize the treatments serials in table-1 with table-3,4 and 6

Mention the area for vermicompost and lime rate

Please solve the problem of t/ha, t ha-1, ton/ha, t ha-1, tons, and ton

Please check the sentence “Table 2. Characteristics the study soil and vermicompost before planting” and correct the sentence

Table-1: Explain, for treatment combination, why don’t use 50% sole lime whether use 50% sole vermicompost?; use 150% sole lime but why not 150% vermicompost? also use combination of 150% VC+100 L but why not 150% L+100% VC

Table-2: BD is appearing very low, whether soil is clay and organic carbon content is very low

Table-3: Check the p vale of available Phosphorus

Table-6: Please use a round figure for NB and MRR, mentioned GNR in the table caption whether used GNB in the table footnote, write correct one

Unify the tables heading with the description heading in the result and discussion section

Please use the same font for footnote

Result and discussion

Please add the field capacity data as irrigation was supplied based on the field capacity

183-184 line: C:N and C/N , write this in a uniform pattern

Please remove clumsiness and duplication during the writing of results and discussion of results. Discuss the results specifically. Omit unnecessary results.

Please reduce the number of references and use recent references, make space for two or more references in the third bracket

Make similarity for writing p, ap, phosphorus, and available phosphorus

Line 264: Place a verb after the reference number 43

Line 276: Put a comma before respectively

Line 303: Remove the bracket before 29.63

Please add the value of the control treatment during the comparison

Partial budget analysis:

First sentence (line 326-329), please read the sentence carefully and rewrite the sentence for better understanding

Line 330: Please use a round figure for data value

Please check the net benefit value described in the abstract, partial budget analysis, and conclusion chapters and make uniform

Conclusion:

Consider the same pattern for RL, LR, and L

Line 344: lime compost? not clear

References:

References were written whimsically. Rewrite the reference chapter and follow a pattern for all reference.

Overall comments:

English of the manuscript is poor. Improvement is required. Please go through many rounds of editing, taking into account typos and grammatical problems, before submitting the updated work.

Following the addition of the aforementioned remarks, the manuscript can be approved.

Reviewer #3: The effort committed to the study is commendable.

However, I have the following observation:

1) There are minor omission of some prepositions.

2) "The file figures could not be opened because there are problems with the contents" was the message I got when I attempted to view the figures. Please re--upload the file figures.

6. PLOS authors have the option to publish the peer review history of their article (what does this mean?). If published, this will include your full peer review and any attached files.

Reviewer #1: No

Reviewer #2: No

Reviewer #3: No

---

## [Author Response · Author response to Decision Letter 0]

7 Aug 2024

The editor raised in point number seven in the journal requirement section it is so difficult to get letter from the source that recommended to use this map/satellite image so better to cancelled this point.

---

## [Decision Letter · Decision Letter 1]

9 Sep 2024

PONE-D-24-13572R1Effects of vermicompost and lime on soil properties and malt barley (Hordeum distichumL.) productivity on acidic soils in Mecha district, northwest EthiopiaPLOS ONE

Dear Dr. Gebreyesus,

Thank you for submitting your manuscript to PLOS ONE. After careful consideration, we feel that it has merit but does not fully meet PLOS ONE’s publication criteria as it currently stands. Therefore, we invite you to submit a revised version of the manuscript that addresses the points raised during the review process.

We look forward to receiving your revised manuscript.

Kind regards,

Ravinder Kumar, Ph.D.

Academic Editor

PLOS ONE

Journal Requirements:

Reviewers' comments:

Reviewer's Responses to Questions

**Comments to the Author**

1. If the authors have adequately addressed your comments raised in a previous round of review and you feel that this manuscript is now acceptable for publication, you may indicate that here to bypass the “Comments to the Author” section, enter your conflict of interest statement in the “Confidential to Editor” section, and submit your "Accept" recommendation.

Reviewer #2: (No Response)

Reviewer #3: (No Response)

2. Is the manuscript technically sound, and do the data support the conclusions?

Reviewer #2: Yes

Reviewer #3: Yes

3. Has the statistical analysis been performed appropriately and rigorously? 

Reviewer #2: Yes

Reviewer #3: Yes

4. Have the authors made all data underlying the findings in their manuscript fully available?

Reviewer #2: Yes

Reviewer #3: Yes

5. Is the manuscript presented in an intelligible fashion and written in standard English?

Reviewer #2: No

Reviewer #3: Yes

6. Review Comments to the Author

Reviewer #2: I am giving thanks to authors for their hard working. But for accept the article they will have to consider the followings

Read the title and reference chapter carefully

Add malt barely in place of integrated in keywords

still problem of old reference in introduction chapter,

still problem of L, RL, t ha-1, t/ha, tons ha-1

Please define it or explain it properly ETB

Please have a careful look the line, 75-77, 127, 133, 190, 247, 251, 254, 269, 275, 303, 333

Remove the clumsiness and duplication of writing in the discussion of soil properties

Please go through many rounds of editing regarding typos and grammatical error

Need improvement of language in result and discussion writing

Reviewer #3: Abstract:

Line 22- please define VC and L at first mention before abbreviation

Line 86- What is NPS? Define first at first mention before abbreviation. Farmers widely apply.... delete used to

Line 108- Define FC, PWP

Line 165- Is there any reference to the economic analysis assumption?

281- Change to (In line with the report of (54),, an increase...

301-What is RL? Let there be consistency in abbreviation. RL are indicated in some part, L in others and Lime elsewhere. Do these represent the same thing? Please review.

332- The application of...

370- 'Provided time' how? If this means permitted the researchers or gave time off to the researchers to conduct the research, please indicate as such so the sentence will nit be ambiguous

7. PLOS authors have the option to publish the peer review history of their article (what does this mean?). If published, this will include your full peer review and any attached files.

Reviewer #2: No

Reviewer #3: No

---

## [Author Response · Author response to Decision Letter 1]

25 Sep 2024

Responses to Editor Comments 

01. Comment: Please review your reference list to ensure it is complete and correct. If you have cited papers that have been retracted, please include the rationale in the manuscript text, or remove these references and replace them with relevant current references. 

01. Response: Thank you for providing constructive and relevant comments. We have carefully reviewed the reference lists and properly revised and corrected them based on the journal reference guidelines. Some old references are removed and replaced with relevant and recent ones. Besides, we have not cited retracted articles in the manuscript. 

Responses to Reviewers' Comments 

Reviewer #1: 

02. Comment: The following research articles need to be cited in the introduction and methodology section before publication:

1. DOI: 10.3390/microorganisms10102078

2. DOI: 10.4236/ojss.2018.82007: 

02. Response: Thank you for your constructive suggestions. We have read both articles thoroughly and cited the articles in the discussion section of the revised manuscript.

Reviewer #2: 

03. Comment: Read the title and reference chapter carefully. 

03. Response: Thank you for your invaluable and constructive comments. We have read the title and reference part critically, and we made significant revisions to the title and references in the revised manuscript accordingly.

04. Comment: Add malt barely in place of integrated with keywords 

04. Response: Thank you so much for your useful feedback and now we have incorporated malt barley instead of integrated in the revised manuscript.

05. Comment: still a problem with the old reference in the introduction chapter 

05. Response: Thank you for your constructive comments. We have removed old references and replaced them with more recent references in the revised manuscript.

06. Comment: still the problem of L, RL, t ha-1, t/ha, tons ha-1

06. Response: Thank you for your insightful observations and relevant comments. We have carefully and consistently rewritten all units and abbreviations in the revised manuscript. 

07. Comment: Please define it or explain it properly ETB

07. Response: Thank you very much for your invaluable comment and we kindly accepted the comment. We have clearly defined ETB as Ethiopian Birr first mentioned before the abbreviation in the revised manuscript.

08. Comment: Please have a careful look the lines, 75-77, 127, 133, 190, 247, 251, 254, 269, 275, 303, 333 

08. Response: Thank you for your constructive and pertinent comments. We have made a careful review of the lines mentioned above and properly rearranged and improved the sentences. 

09. Comment: Remove the clumsiness and duplication of writing in the discussion of soil properties 

09. Response: Thank you very much for your invaluable comments. We have carefully read the manuscript and removed unnecessary literature and duplication of writing in the discussion of soil properties in the revised manuscript

10. Comment: Please go through many rounds of editing regarding typos and grammatical error 

10. Response: Thank you for reading of the manuscript in-depth and giving useful comments. Thus, we have read the manuscript carefully and properly addressed the typos and grammatical errors.

11. Comment: Need improvement of language in result and discussion writing 

11. Response: Thank you very much for your pertinent comments and feedback. Now we have made significant language improvements in result and discussion writing. 

Reviewer #3: 

12. Comment: Line 22- please define VC and L at first mention before the abbreviation

12 Response: Thank you for your constructive and insightful comments. We have properly defined VC as vermicompost and L as lime as first mentioned in the revised manuscript.

13. Comment: Line 86- What is NPS? Define first at first mention before abbreviation. Farmers widely apply.... delete used to

13. Response: Thank you very much for your invaluable comments. We have defined NPS as a Nitrogen-Phosphorus-Sulfur blend fertilizer at first mention before the abbreviation. Besides, we deleted the words used in the revised manuscript accordingly. 

14. Comment: Line 108- Define FC, PWP

14. Response: Thank you for your critical observations and for giving pertinent feedback. Now we have defined FC(field capacity) and PWP (permanent wilting point) in the revised manuscript.

15. Comment: Line 165- Is there any reference to the economic analysis assumption?

15. Response: Thank you for raising invaluable comments. Yes, there is a reference to the economic analysis assumption. The assumption stated that grain and straw yields produced from research sites need to be adjusted down by 10% to narrow the yield gap between yields obtained from experimental plots and farmers’ fields. Hence, economic analysis was done based on the method provided by CIMMYT (1988).

16. Comment: Line 281- Change to In line with the report of (54), an increase...

16. Response: Thank you for your useful comments and we have made corrections in the revised manuscript accordingly. 

17. Comment: Line 301-What is RL? Let there be consistency in abbreviation. RL are indicated in some parts, L in others, and Lime elsewhere. Do these represent the same thing? Please review. 

17. Response: Thank you for your invaluable and constructive comments. We kindly accepted the comment, and now we have used L (lime) instead of RL(recommended lime) throughout the revised manuscript.

18. Comment: Line 332- The application of... 

18. Response: Thank you for giving pertinent comments.. We have properly rewritten the sentence in the revised manuscript. 

19. Comment: Line 370- 'Provided time' how? If this means permitting the researchers or giving time off to the researchers to conduct the research, please indicate as such so the sentence will not be ambiguous.

19. Response: Thank you very much for your insightfull observation and feedback. The meaning of this sentence is that I thank the Amhara Regional Bureau of Agriculture and Woreta College of Agriculture for permitting me to pursue my Ph.D. studies. Now we have rearranged and improved this sentence in the revised manuscript

---

## [Editor Report · Decision Letter 2]

27 Sep 2024

Effects of Vermicompost and Lime on Acidic Soil Properties and Malt Barley (Hordeum DistichumL.) Productivity in Mecha District, Northwest Ethiopia

PONE-D-24-13572R2

Dear Dr. Gebreyesus,

We’re pleased to inform you that your manuscript has been judged scientifically suitable for publication and will be formally accepted for publication once it meets all outstanding technical requirements.

Kind regards,

Ravinder Kumar, Ph.D.

Academic Editor

PLOS ONE

---

## [Editor Report · Acceptance letter]

9 Oct 2024

PONE-D-24-13572R2 

PLOS ONE

Dear Dr. Gebreyesus, 

I'm pleased to inform you that your manuscript has been deemed suitable for publication in PLOS ONE. Congratulations! Your manuscript is now being handed over to our production team.

Kind regards, 

on behalf of

Dr. Ravinder Kumar 

Academic Editor

PLOS ONE